



# Molecular composition and photochemical lifetimes of brown carbon chromophores in biomass burning organic aerosol

Lauren T. Fleming,[1] Peng Lin,[2,*] James M. Roberts,[3] Vanessa Selimovic,[4] Robert Yokelson,[4] Julia Laskin,[2] Alexander Laskin,[2] Sergey A. Nizkorodov[1]

[1]Department of Chemistry, University of California, Irvine, Irvine, CA 92697, USA
[2]Department of Chemistry, Purdue University, West Lafayette, IN 47907, USA
[3]Chemical Sciences Division, Earth System Research Laboratory, National Oceanic and Atmospheric Administration, Boulder, CO 80305, USA
[4]Department of Chemistry, University of Montana, Missoula, 59812, USA
[*]Now at California Air Resources Board, El Monte, CA, 91731

*Correspondence to:* Sergey A. Nizkorodov (nizkorod@uci.edu)

**Abstract.** To better understand the effects of wildfires on air quality and climate, it is important to assess the occurrence of chromophoric compounds in smoke and characterize their optical properties. This study explores the molecular composition of light-absorbing organic aerosol, or brown carbon (BrC), sampled at the Missoula Fire Sciences laboratory as a part of the FIREX Fall 2016 lab intensive. Twelve biomass fuels from different plant types were tested, including gymnosperm (coniferous) and angiosperm (flowering) plants, and different ecosystem components such as duff, litter, and canopy. Emitted particles were collected onto Teflon filters and analyzed offline using high performance liquid chromatography/ photodiode array/ high resolution mass spectrometry (HPLC/PDA/HRMS). Separated BrC chromophores were classified by their retention times, absorption spectra, integrated PDA absorbance in the near-UV and visible spectral range (300-700 nm), and chemical formulas from the accurate m/z measurements. BrC chromophores were grouped into the following classes and subclasses: lignin-derived, which includes lignin pyrolysis products; distillation products, which include coumarins and flavonoids; nitroaromatics; and polycyclic aromatic hydrocarbons (PAHs). The observed classes/subclasses were common across most fuel types, although specific BrC chromophores varied based on plant type (gymnosperm or angiosperm) and ecosystem component(s) burned. To study the stability of the observed BrC compounds with respect to photodegradation, biomass burning organic aerosol (BBOA) particle samples were irradiated directly on filters with near UV (300-400 nm) radiation, followed by extraction and the HPLC/PDA/HRMS analysis. Lifetimes of individual BrC chromophores depended on the fuel type and the corresponding combustion conditions, but lignin-derived and flavonoid classes of BrC generally had the longest lifetimes with respect to UV photodegradation. Moreover, lifetimes for the same type of BrC chromophores varied depending on biomass fuel and combustion conditions. While individual BrC chromophores disappeared on a timescale of several days, the overall light absorption by the sample persisted longer, presumably because the photolysis processes converted one set of chromophores into another without complete photobleaching, or from undetected BrC



chromophores that photobleached more slowly. To model the effect of BrC on climate, it is important to understand the change in the absorption coefficient with time. We measured the equivalent atmospheric lifetimes of the overall BrC absorption coefficient which ranged from 10 to 41 days, with subalpine fir having the shortest lifetime, and conifer canopies having the longest. BrC emitted from biomass fuel loads encompassing multiple ecosystem components (litter, shrub, canopy) had absorption lifetimes on the lower end of the range. These results indicate that photobleaching by atmospheric

photolysis is relatively slow. Other chemical aging mechanisms, such as heterogeneous oxidation by OH, may be more important for BrC degradation than photolysis for predicting the decay of BBOA BrC absorption in models.

## 1 Introduction

Forests have naturally occurring wildfire cycles that maintain the forest ecosystem, but global climate change is altering the

45 cycles with unpredictable consequences (Shvidenko and Schepaschenko, 2013; Weber and Stocks, 1998). In addition to the impacts of wildfires on ecosystems, biomass burning plumes have pronounced effects on atmospheric chemistry and climate. Wildfires continue to increase in frequency and intensity worldwide with climate change (Boulanger et al., 2014; Moriondo et al., 2006; Shvidenko and Schepaschenko, 2013; Wotton et al., 2010; Wotton and Flannigan, 1993). Wildfire plumes contain a complex mixture of greenhouse gases (carbon dioxide and methane), multiple non-methane organic compounds

(NMOCs), and carbonaceous and ash particles. The effects arising from biomass burning organic aerosol (BBOA) particles are not well understood because BBOA composition and optical properties may depend on many factors, such as the type of fuel burned and combustion conditions (Chen and Bond, 2010; Jen et al., 2019; Kirchstetter et al., 2004), wind speed, heading or backing fires (Surawski et al., 2015), and fuel moisture content (Tihay-Felicelli et al., 2017). Global climate models are starting to include contributions from light-absorbing organic carbon, termed brown carbon (BrC), because

treating BBOA particles as purely scattering leads to incorrect predictions of climate forcing (Bond et al., 2011; Laskin et al., 2015; Ma et al., 2012). One of the first studies incorporating BrC into models was by Feng et al. (2013), which found that in areas where primary BrC emissions are high, the absorbing component of BBOA particles can dominate over the scattering component, switching net radiative forcing by organic aerosols from negative (cooling) to positive (warming) at the top of the atmosphere. Other modeling studies have demonstrated that BrC can have large effects on the radiative forcing (Bahadur

et al., 2012; Chung et al., 2012; Laskin et al., 2015; Ramanathan et al., 2007). However, field measurements to date indicate that BrC has a short lifetime of ~10 hours, which would considerably reduce its impact if included in models (Forrister et al., 2015; Selimovic et al., 2019). Light-absorption by BrC can also result in a significant decrease in the photolysis rates of photochemically active gases, such as HONO and nitrogen dioxide, which affect the mixing ratios of ozone and hydroxyl radicals (Jiang et al., 2012; Li et al., 2011). To better quantify their effect on climate, lifetimes, light absorption properties

and chemical composition of a broader range of BrC and BBOA particles should be studied.



Previous studies have identified important classes of BBOA chemical components that are responsible for light absorption. A major class includes lignin-pyrolysis products, which are typically substituted aromatics with a high degree of unsaturation, such as coniferaldehyde (Budisulistiorini et al., 2017; Fleming et al., 2018; Simoneit et al., 1993). Another class is nitroaromatics, such as nitrocatechols, which are readily produced in the presence of NO$_x$ and absorb strongly with a $\lambda_{max}$ around 340 nm (Iinuma et al., 2010; Lin et al., 2017). Polyclyclic aromatic compounds (PAHs) have long been known to be emitted from incomplete combustion processes, and large PAHs can be significantly light-absorbing at the near-UV and visible wavelengths (Simoneit, 2002). Budisulistiorini et al. (2017) observed sulfur-containing species from fern and peat pyrolysis, and suggested that they are formed via acid-catalyzed heterogeneous reactions. Tar balls are largely externally-mixed spheres or spherical aggregates produced from smoldering combustion or through multi-phase secondary chemistry (Sedlacek III et al., 2018; Tóth et al., 2014). In terms of their chemical composition, tar balls are thought be comprised primarily of oxygenated organic compounds, similar to that of BBOA particles (Chakrabarty et al., 2010; Girotto et al., 2018; Li et al., 2019; Pósfai et al., 2004; Sedlacek III et al., 2018).

BrC components undergo photochemical transformations during atmospheric transport, including photobleaching or photoenhancement of their absorption coefficients. For example, the field studies of Forrister et al. (2015) and Selimovic et al. (2019) observed a substantial decay in aerosol UV light absorption in biomass burning plumes corresponding to a half-life of 9 to 15 hours. Similarly, Lin et al. (2017) reported rapid evolution of both the BBOA composition and optical properties during a nationwide biomass burning event in Israel. These changes in BBOA properties are supported by laboratory studies of photochemical aging of BBOA particles or relevant surrogates via heterogeneous photooxidation (exposure of particles to gaseous OH), aqueous photooxidation (exposure of BBOA compounds to OH within cloud/fog water droplets), direct photolysis (exposure of particles or their aqueous extracts to actinic UV radiation), and indirect photolysis (photosensitized reactions between BBOA molecules and electronically-excited triplet states of photosensitizers). Several studies have characterized changes in the UV-Vis spectra of nitrophenols, common in BBOA particles, as they are exposed to UV radiation. For example, Hinks et al. (2016) irradiated 2,4-dinitrophenol incorporated in limonene secondary organic aerosol material and observed the absorbance decrease in the range of 250-320 nm, while the absorbance from 400-450 nm increased. Similarly, Zhao et al. (2015) observed a photoenhancement at 420 nm for a 4-nitrocatechol aqueous solution in response to direct photolysis. During photooxidation with OH (produced by an intentional addition of hydrogen peroxide to the photolyzed solution), photoenhancement at 420 nm was observed initially, but the solution photobleached within an hour. In Hems and Abbatt (2018), aqueous solutions of nitrophenols and hydrogen peroxide were irradiated, atomized, and then analyzed by an aerosol-CIMS. This study attributed the photoenhancement at 420 nm to the functionalization of nitrophenols, followed by their photodegradation at 420 nm as was evidenced by fragmentation of functionalized nitrophenols. Lignin pyrolysis products and other lignin-derived molecules have been shown to be oxidized into light-absorbing compounds under certain conditions. For example, Gelencsér et al. (2003) observed an increase in absorption at visible wavelengths during the photooxidation of single component aromatic hydroxyl acids in aqueous solutions. Chang and



Thompson (2010) and Tang and Thompson (2012) observed production of light absorbing compounds during aqueous reactions of OH with multiple phenolic compounds. Smith et al. (2016) found that triplet-excited molecules can react with phenolic compounds in cloud water mimics producing BrC chromophoric products. In Kaur et al. (2019), five model BBOA model compounds were irradiated and hydroxyl radicals, singlet molecular oxygen, and triplet excited state molecules were detected with probe molecules. They found that all model compounds decayed on the order of hours from indirect photooxidation. There are many studies that have investigated the photodegradation of PAHs on ice surfaces, ocean water mimics, and soil (Smol and Włodarczyk-Makuła, 2017). The degradation of the three-ring PAH phenanthrene had a half-life of 13 to 23 hours depending on the solvent it was dissolved in (Shankar et al., 2019). Using infrared spectroscopy they observed the emergence of carboxylic acid, aldehyde, and ketone functionalities during photolysis. Miller and Olejnik, 2001 irradiated aqueous solutions of PAH mixtures with UVC lamps. They found that the photodegradation of benzo[a]pyrene and chrysene proceeds more rapidly at acidic pH values and proposed a mechanism based on their findings (Miller and Olejnik, 2001).

The photochemical aging of actual mixtures of BBOA compounds, not just surrogates, was also reported in the literature. For example, Tomaz et al. (2018) found that aqueous BBOA mixtures from the 2016 FIREX lab intensive decayed rapidly, with most having lifetimes due to aqueous OH oxidation mimicking clouds of a half a day or less. The decay of compounds such as catechol, benzoic acid, and methylfurfural lead to the formation of oxalate, which made up 13-16% of total dissolved organic carbon after 150 hours. Saleh et al. (2013) burned pocosin pine and oak, and diluted smoke was irradiated with UV lights in a smog chamber. Aerosol optical properties were monitored with an aethalometer at seven wavelengths. They found that aged emissions were more absorbing than fresh emissions at 370 and 470 nm after one hour. Zhong and Jang, (2014) tracked the absorption coefficients of BBOA during solar exposure in a smog chamber, and observed an increase of 11-54% in the integrated mass absorption cross section (280-600 nm) in the first half of the day followed by decrease in the afternoon. In Lin et al. (2016), BBOA particles collected from ponderosa pine and Indonesian peat burns were dissolved in a 50% by volume water/acetonitrile solvent and irradiated with actinic wavelengths. They found that regardless of the fuel type the half-life of the absorbance at 300 nm was roughly 16 hours under sunlight for soluble BBOA particles. Wong et al. (2017) found that irradiated BBOA water extracts lost water soluble organic carbon (WSOC). Simultaneously, the absorption coefficients at 365 nm and 400 nm first increased, in the latter case to about 0.035 $m^2$ $g^{-1}$ after 20 minutes of illumination time, and then decreased to nearly zero in 60 minutes. In Sumlin et al. (2017), BBOA particles produced from Alaskan peat were aged by reactions with OH in an oxidation flow reactor (OFR), and light absorption coefficients of aged and unaged BBOA particles were measured by an integrated photoacoustic nephelometer. They found that the mass absorption coefficient at 375 nm deceased roughly 45% after an equivalent of 4.5 days of photochemical aging.

To summarize the brief literature survey above, much work has been done in terms of characterizing optical properties of photochemically aged BBOA particles and surrogates, but a consensus on what drives the photobleaching and photoenhancement of chromophores in BBOA and the relative importance of these processes on atmospherically relevant





time scales has not been reached. This study aims to better understand the molecular composition of BrC for different fuel types and combustion conditions as it may be essential to understanding the optical properties of BBOA and predicting their evolution during photochemical aging.

This study explored the diversity in the molecular composition of BrC chromophores found in BBOA samples generated by burning forest fire fuels, and examined how the chemical composition and optical properties change during UV irradiation of BrC materials in the absence of gas-phase oxidants. BBOA samples from twelve biomass fuels collected from around the United States, encompassing both gymnosperm and angiosperm plant types and different parts of the ecosystem; including duff, litter, and canopy were examined. Samples collected on filters were extracted by solvents with a range of polarities and

analyzed by high performance liquid chromatography coupled to a photodiode array and high resolution mass spectrometer (HPLC/PDA/HRMS) to target BrC chromophores. To investigate whether the BrC chromophores are photolabile or photostable, BBOA particles samples were directly irradiated on filter substrates before analysis by HPLC/PDA/HRMS or UV-Vis spectrometry. We estimated their photolysis lifetimes in BBOA particles by measuring the time resolved absorbance of individual chromophores. We found that the equivalent atmospheric lifetime due to photolysis of individual chromophores

ranged from 0.4-1.6 days, which is a relevant timescale for long-range atmospheric transport. BrC chromophores could survive the exposure to UV radiation on different timescales, depending on their molecular structure or their interactions with neighboring molecules dictated by BBOA type. However, the overall absorption by BrC (integrated over 300-700 nm) persisted longer due to direct photolysis, with lifetimes ranging from 10-41 days, because some photolysis products of the original BrC are also light-absorbing. The equivalent atmospheric photolysis lifetimes of BrC absorption are long compared

to typical lifetimes for heterogeneous oxidation of BBOA particles by OH. For climate modeling applications these results suggest that chemical aging mechanisms other than photolysis may play a more significant role in the evolution of BrC properties.

## 2 Experimental methods

### 2.1 *Sample collection and information*

BBOA particle samples were collected at the FIREX Fall 2016 lab intensive at the Missoula Fire Lab (https://www.esrl.noaa.gov/csd/projects/firex/firelab/). One of the BBOA samples used in this study was from a "stack" burn and the other samples were from "room" burns. Selimovic et al. (2018) explains room and stack burns and fuels in detail. Briefly, the combustion of forest fire fuels lasted 5-20 minutes and during stack burns emissions were collected from a

160 constant, diluted flow of entrained emissions by way of the stack. In room burns, the smoke from the fire was allowed to mix in the room during sample collection, and BBOA was collected during both the burn and mixing periods. Smoke was purged from the room by clean air in between burns. Fuels were collected from different US regions and brought to the Missoula



Fire Lab for test burns. This manuscript focuses on twelve fires covering gymnosperm or conifers, including ponderosa pine (*Pinus ponderosa*), lodgepole pine (*Pinus contorta*), Engelmann spruce (*Picea engelmanii*), Douglas fir (*Pseudotsuga*
*menziesii*), juniper (*Juniperus*), longleaf pine (*Pinus palustris*), rotten log, and subalpine fir (*Abies lasiocarpa*). Angiosperm forest fire fuels included Montana sagebrush and two types of chaparral, manzanita (*Arctostaphylos*) and chamise (*Adenostoma fasciculatum*). In some test burns, a representative "ecosystem" mix of biomass was used, including canopy, duff, litter, herbaceous, and shrub components. In other test burns, single biomass components of the ecosystem were used. Information for each fire is provided in Table S1.

Copper tubing with a $PM_{2.5}$ cyclone inlet was placed in the combustion room while the pump and filter were located in an adjacent room. The pump was operating at a flow of 16.7 L/min with the aid of a critical orifice, and BBOA particle samples were collected on PTFE filter substrates (FGLP04700, Millipore, 47 mm diameter, 0.2 µm pore size) during both of the combustion and smoke mixing stages of the room burns. Loaded filters were stored at -18°C until they were analyzed. The room burn protocols allowed for long collection times and therefore higher aerosol mass loading, which is desirable for the
analysis described below.

### 2.2 HPLC/PDA/HRMS

The molecular identity and relative abundance of BrC chromophores were determined using the High Performance Liquid Chromatography/Photodiode Array/High Resolution Mass Spectrometry (HPLC/PDA/HRMS) platform (Fleming et al., 2018; Lin et al., 2018). Segments of the filter were extracted into a mixture of organic solvents composed of 2.0 mL
dichloromethane, 2.0 mL acetonitrile, and 1.0 mL of hexanes. The extraction occurred overnight on a platform shaker. Extracts were filtered with PVDF syringe filters (Millipore, Duropore, 13 mm, 0.22 µm) to remove undissolved suspended particles. Water (50 µL) and DMSO (100 µL) were added to the extracts, which were then concentrated under a flow of $N_2$ until the volume was reduced to roughly 150 µL, which signified that the extracting solvent evaporated and (mostly) water and DMSO remained in the solution. For photolyzed BBOA particles, DMSO (30 µL) was exclusively added to the extract,
and evaporated to a volume of 30 µL.

The HPLC utilized a reverse-phase column (Luna C18, 2 × 150 mm, 5 µm particles, 100 Å pores, Phenomenex, Inc.). The injection volume was 5.0 µL for unphotolyzed or 10 µL for extractions of post-photolysis samples, with the latter providing more analyte mass since only a quarter of the filter was used in photolysis experiments. The mobile phase consisted of 0.05 % formic acid in LC–MS grade water (A) and LC–MS grade acetonitrile (B). Gradient elution was performed with the A–B
mixture at a flow rate of 200 µL min$^{-1}$: 0–3 min hold at 90 % A, 3– 62 min linear gradient to 10 % A, 63–75 min hold at 10 % A, 76–89 min linear gradient to 0 % A, 90–100 min hold at 0 % A, then 101–120 min hold at 90 % A. The electrospray ionization (ESI) settings of the Orbitrap HRMS were as follows: 4.0 kV spray potential, 35 units of sheath gas flow, 10 units





of auxiliary gas flow, and 8 units of sweep gas flow. The solutions were analyzed in both positive and negative ion ESI-HRMS modes.

The HPLC/PDA/HRMS data were acquired and first analyzed using Xcalibur 2.4 software (Thermo Scientific). Possible exact masses were identified based on the corresponding LC retention time using the open source software toolbox MZmine version 2.23 (http://mzmine.github.io/) (Pluskal et al., 2010). Chemical formulas were assigned from exact m/z values using the Formula Calculator v1.1. More details about experimental procedures and data processing can be found elsewhere (Lin et al., 2015b, 2016, 2018).

**2.3 Condensed-phase photolysis**

A quarter of the filter was directly irradiated by either an ultraviolet light-emitting diode (LED, Thorlabs M300L4) or a filtered Xenon arc lamp. The LED was used in experiments aimed at estimating lifetimes of individual chromophores. The LED emission spectrum was centered at 300 nm with a FWHM of 20 nm. This wavelength was chosen because it corresponds to the most energetic UV photons available in the lower troposphere. It is a common practice in photochemical
experiments to use narrow band UV sources, as opposed to a broadband simulator, as it limits sample heating and evaporation (Calvert and Pitts, 1966). The LED was fixed half a centimeter away from the filter resulting in an incident power density of 11 mW cm$^{-2}$. Irradiation times for these experiments are given in Table 3. After the irradiation step, the photolyzed BBOA particles were extracted and analyzed using HPLC/PDA/HRMS as described in the previous section.

The irradiation time by the LED was converted into an equivalent time under sunlight by calculating the ratio of the 290-350
210   nm integrated spectral flux of the sun and the 300 nm LED, given in equation 1. This conversion assumes that photochemistry is limited to the < 350 nm range, consistent with the photochemistry of many organic molecules, which exhibit a sharp drop in the photochemical quantum yields at longer wavelengths (Turro et al., 2009). Because the radiation source does not replicate the solar spectrum, the lifetimes calculated from the formula below should be regarded as estimates.

$$\tau_{atm} = \tau_{LED} \times \frac{\int_{290nm}^{350nm} F_{LED}(\lambda) d\lambda}{\int_{290nm}^{350nm} \langle F_{solar}(\lambda) \rangle_{24hr} d\lambda}$$
(1)

The spectral flux density for the LED and the sun as a function of wavelength is shown in Figure 1. The solar flux density was estimated every hour and averaged over a 24-hour period for Los Angeles, CA (34°N 118°W) on June 20, 2017 from the quick TUV calculator (Madronich et al., 2002), using the following parameters: 300 du overhead ozone column, 0.1 surface albedo (0-1), and ground elevation of 0 km with default outputs for aerosols and clouds. The procedure for calculating the
spectral flux density of the LED is described in the supporting information. The maximum possible spectral flux density





from the sun was also calculated at a solar zenith angle (SZA) of 0° using the TUV calculator. The equation for calculating the equivalent atmospheric lifetime at an SZA of 0° is the same as equation (1) except that the 24-hour averaged flux density is replaced by the peak flux density at SZA = 0. The SZA=0° comparison represents the lower limit of BrC absorption lifetimes.

In a separate series of experiments, filter samples were irradiated by the filtered radiation from a xenon arc lamp to determine the characteristic lifetime for the photobleaching of the overall absorption by BrC molecules. A quarter of a PTFE filter sample was exposed to filtered light emitted from a xenon-arc lamp (Newport 66902). Broadband light was reflected at a 90° angle using a dichroic mirror, then filtered through a 295 nm long-pass filter (Schott WG295), and finally passed through a UV bandpass filter (Schott BG1) ultimately transmitting light in the range of 290-400 nm. The incident overall
power density was 196 mW/cm$^2$. Particles were irradiated for ~12 hours to 1.8 days; the exact time varied from sample to sample depending on the offline transmission spectra. Transmission spectra were acquired directly from the PTFE filter without any material extraction using a Jasco V-670 absorption spectrometer, with a blank PTFE filter used as a reference. Four to six transmission spectra were collected at each time point as the filter was rotated, to minimize the effect of the filter orientation. The filter was then returned to the photolysis set up for further irradiation. When there was no longer any change
in the transmission spectrum due to irradiation, the filter was extracted into an organic solvent mixture of 10 mL methanol, 5.0 mL acetonitrile, and 2.0 mL of hexane in a scintillation vial using a vortex mixer. The solution was then evaporated in order to increase the analyte concentration. For comparison, an un-irradiated quarter of the filter was prepared identically in a separate vial, and solution-phase transmission spectra of both solutions were recorded using a dual beam UV-Vis spectrometer (Shimadzu UV-2450). Sample filter-based and solution phase spectra are shown in Figure S2, with the Y-axis
converted to effective base-10 absorbance, A = -log(T), where T is the wavelength-dependent transmittance through the filter or the cuvette. For filter-based transmission spectra, the baseline was manually corrected by assuming the absorbance at 850 nm was 0 for BrC.

In all photolysis experiments, the integrated absorbance from 300 to 700 nm was calculated and normalized to the un-irradiated absorbance. The decay constants and corresponding lifetimes were calculated as described in Figure S1. The linear
regression trend line was constrained to have a y-intercept of zero, since this represents the log of the un-irradiated absorbance normalized to itself.

**3 Results and Discussion**

**3.1 BrC Chromophores**

Table 1 summarizes BrC chromophores observed in two or more fires or fuel types. The table numbers BrC chromophores
by their ascending retention time on the HPLC column, i.e., with smaller, more polar compounds appearing first. Each entry includes the absorption spectrum recorded by the PDA detector, the chemical formula/s corresponding to the detected





characteristic masses at that retention time, and a potential structure based on a reference spectrum or observations in previous studies. All PDA chromatograms were integrated over 300-700 nm and normalized to the maximum integrated absorbance. Chromophores in Table 1 are binned with respect to their normalized PDA absorbance as M-Major (75%-100%), I-Intermediate (25%-75%), or W-Weak (5%-25%). Compounds making up less than 5% of the normalized absorbance are not included in the tables.

Lignin pyrolysis products make up one group of BrC chromophores observed. Lignin is a large, heterogeneous biopolymer that is a significant component of wood, along with cellulose and hemicellulose. Lignin monomer units vary depending on the class of the plant, but generally possess phenolic moieties that are largely preserved during pyrolysis (Simoneit et al., 1993). Sinapaldehyde (8) and coniferaldehyde (9) are known lignin pyrolysis products derived from the corresponding lignin monomer units, sinapyl and coniferyl alcohol, respectively. However, they are detected in varying abundance depending on the lignin monomer units of the plant class. Sinapaldehyde and coniferaldehyde are separated by the column, but elute only 0.3 minutes apart as shown in Figure 2. Sinapaldehyde is a major BrC chromophore for nearly all angiosperm or flowering fuel types, including ceanothus, chamise, and sagebrush, while coniferaldehyde is a major BrC chromophore largely among conifers or soft wood species such as subalpine fir duff, longleaf pine, juniper, and ponderosa pine litter. Coniferaldehyde has one less methoxy ring substituent compared to sinapaldehyde, and its PDA intensity is generally anti-correlated to that of sinapaldehyde. In other words, for fuel types with low sinapaldehyde absorbance, we observe coniferaldehyde as a major BrC chromophore and vice versa. This is consistent with the composition of lignin monomers for angiosperms and gymnosperms (Sarkanen and Ludwig, 1971; Simoneit et al., 1993).

Other BrC chromophores cannot be classified as lignin pyrolysis products but are clearly lignin-derived. Vanillic acid (1) elutes at 10.07-10.29 minutes as the first, shared chromophore across multiple fuel types that is notable in terms of absorption. It is observed in three fires as a weak chromophore, including subalpine fir duff, ponderosa pine rotten log, and Engelmann spruce duff. All three fires are dominated by smoldering combustion and have the lowest modified combustion efficiencies (MCEs) of all fires (Table S1). This evidence suggests that vanillic acid is a product of smoldering combustion. Further, it also has the coniferyl moiety observed for softwoods. Salicylic acid (3) is an intermediate absorbing BrC chromophore produced during lodgepole pine burning, and weakly absorbing among other softwoods and duffs. Veratraldehyde (4) is another lignin-derived BrC chromophore, which appears in nearly all BBOA samples of this study, regardless of whether they are gymnosperm or angiosperm fuels.

There are other BrC chromophores with $C_xH_yO_z$ composition that can be explained as distillation products, or the volatilization of molecules originating in plants as secondary metabolites (Agati et al., 2012; Iranshahi et al., 2009). Found in plants, coumarins such as umbelliferone (5) and nodakenetin (13) have been researched because of their positive pharmacological properties (Venugopala et al., 2013). The absorption spectrum for nodakenetin has not been reported, however, the molecule has previously been detected in plant tissues (Lee et al., 2003; Wang et al., 2014), and is a



major/intermediate BrC chromophore in smoke from all fuel types except chamise and ceanothus. Another type of distillation product is flavonoids, which give leaves, flowers, and fruits their color protecting the plant from solar UV radiation, and are antioxidants-- guarding the plant from reactive oxygen species (Agati et al., 2012). Flavones and flavonols have the backbone structure of 2-phenyl-1-benzopyran-4-one, and flavonols additionally require a hydroxy substituent on the only available carbon of the pyranone ring. BrC chromophores 11, 14, and 16 could have flavonoid structures based on their chemical formulas. Interestingly, tentatively assigned kaempferol (11) and diosmetin (14) are observed in only conifer species, such as lodgepole pine and longleaf pine. On the other hand, 7-hydroxy-3',4'-dimethoxyflavone (16) is only observed in angiosperm BBOA particles: ceanothus, chamise, and sagebrush. The former two plants appear to be related as they have the order *rosales* in common, which could explain the same flavone detected in both. Coumarins and flavonoids were distillation products observed across fuel types, although the observation of specific BrC chromophores depends on the plant class, angiosperm or gymnosperm.

Nitroaromatics are a strongly-absorbing class of BrC chromophores that are formed from the reaction of aromatics with $NO_x$ in plumes (Harrison et al., 2005). This class of compounds is represented in Table 1 with nitropyrogallol (2), nitrocatechol (6), hydroxynitroguaiacol (7), and methyl nitrocatechol (10). Xie et al. 2019 suggests that chromophore (12) with the chemical formula $C_{11}H_{13}NO_5$ is not a nitroaromatic compound, but rather, a compound containing a different nitrogen-containing functional group, such as a nitrile group. We did not observe this group of chromophores for fires with low $NO_x$ levels, such as duff, as qualitatively indicated by the peak NO level (Table S1). Nitrocatechol and methyl-nitrocatechol are tracers for BBOA emissions formed from the photooxidation of phenol or *m*-cresol, toluene and other aromatic compounds in the presence of $NO_x$ (Iinuma et al., 2010, 2016; Lin et al., 2015a). These chromophores are most prominent in BBOA particles from chamise and sagebrush burns. Those two fires exhibited the highest NO mixing ratios in the entire study– 3.79 ppmv (82% of total N emissions) and 1.62 ppmv (57% of total N emissions) peak NO values, respectively. Nitropyrogallol (2) has an additional hydroxy group and is likely formed in the same way as nitrocatechol and methyl nitrocatechol, but is more oxidized. A compound with the same formula as nitropyrogallol (2) was observed during the photooxidation of nitrocatechol in the lab (Hems and Abbatt, 2018). This is an intermediate or major BrC chromophore detected in BBOA samples from longleaf pine, manzanita, and ponderosa pine litter fires. Hydroxynitroguaiacol (7) was observed in 10 of the 12 fires, and is most prominent in ponderosa pine log BBOA particles despite this fire having the lowest NO levels. However, it may still form through photooxidation of guaiacol in the presence of $NO_x$ (Hems and Abbatt, 2018). Nitrocatechol and methyl nitrocatechol are often used as biomass burning tracers in aged plumes (Al-Naiema and Stone, 2017; Iinuma et al., 2010; Li et al., 2016). However in addition to these, we observed more oxidized versions of these nitroaromatic species with varying abundance depending on the BrC chromophore and test fire. This suggests that the BBOA markers nitrocatechol and methyl nitrocatechol become more functionalized on relatively short time scales (less than two hours) due to photooxidative aging.





Polycyclic aromatic hydrocarbons (PAHs) are known to be products of incomplete combustion, and they have the potential to be long-lived BrC chromophores. PAHs have been observed in pristine environments, suggesting this class of BrC chromophores is stable during atmospheric transport (Fernández et al., 2002; Macdonald et al., 2000; Sofowote et al., 2011; Zhou et al., 2012). In addition to its climatic effects, PAHs are mutagenic and carcinogenic as their metabolites, diol
epoxides, bind to guanidine nucleobases in DNA effectively leading to mutations (Finlayson-Pitts and Pitts, 2000; Moorthy et al., 2015; Wood et al., 1984; Xue and Warshawsky, 2005; Zhou et al., 2017). Various PAHs (17-25, Table 1) were observed in only ceanothus, chamise, and sagebrush BBOA particles. PAHs in Table 1 are detected from positive ion mode ESI, and although positive mode ESI is not optimal for observing PAHs, larger PAHs are detectable by this method (Cha et al., 2018). The same PAHs were previously observed by Lin et al., 2018 for sagebrush using atmospheric pressure
photoionization (APPI) coupled with HPLC/PDA/HRMS, which is more sensitive for the detection of non-polar aromatic compounds. In general, individual PAH chromophores are binned as "weak" in Table 1 based on their contribution to optical absorption, but for BBOA sampled from flaming sagebrush and chamise burns, they make up a significant fraction of the overall light-absorption by BrC.

Table 2 presents abundant BrC chromophores observed only in a single type of biomass fuel emissions. It should be noted
that compounds making up less than 5% of the normalized PDA absorbance (integrated from 300-700 nm) are not included in the tables. Due to this constraint, chromophores in Table 2 may also be present in other fires, but at very low PDA absorbance values. Despite BrC chromophores in Table 2 being observed significantly for only one fuel type, they belong to the same compound classes as the BrC chromophores in Table 1. For example, a coumarin known as scopoletin (26) was observed from sagebrush BBOA. Previously we discussed these coumarins are possible distillation products, along with
flavonoids, which we also observe as a product (40) from the ceanothus fire. These distillation products (26 and 40) are among the most strongly absorbing of the BrC chromophores, characterized as intermediate or "I" in Table 2.

### 3.2 Aging by condensed-phase photolysis

Gymnosperm (lodgepole pine) and angiosperm (ceanothus) BBOA particle samples were selected for the initial condensed-
phase photolysis experiments. BBOA filter samples from a lodgepole pine burn were photolyzed for 6 hours by an LED centered around 300 nm (which corresponds to approximately 33 hours of photochemical aging from 24-hour average spectral flux density, see equation 1). BBOA particles from the ceanothus burn were photolyzed by the same LED for 16 hours (equivalent to 88 hours of 24 hour averaged atmospheric sunlight). The burning of gymnosperm (lodgepole pine) and angiosperm (ceanothus) resulted in different distributions of BrC chromophore classes. However the same compound
classes, lignin-derived and flavonoid compounds, were photo-resistant in both samples.





Most chromophores from the lodgepole pine burn sample experienced complete photobleaching during this exposure, but six of them remained observable, including coniferaldehyde ($C_{10}H_{10}O_3$, 80% decrease), salicylic acid ($C_7H_6O_3$, 70% decrease), veratraldehyde ($C_9H_8O_3$, 90% decrease), flavonoids ($C_{15}H_{10}O_6$ & $C_{16}H_{12}O_6$, both 70% decrease), and nodakenetin ($C_{14}H_{14}O_4$, 90% decrease), as shown in Figure 3. Figure 4 shows five chromophores from the ceanothus burn sample that remain
observable under these conditions including sinapaldehyde ($C_{11}H_{12}O_4$, 90% decrease), a lignin-derived chromophore ($C_{18}H_{16}O_6$, 80% decrease), and flavonoids ($C_{16}H_{12}O_5$, $C_{17}H_{14}O_6$, and $C_{17}H_{14}O_5$, all 80% decrease), some of which are observed exclusively in this fire. These comparatively resilient species are aromatic, which helps them be more resistant to photodegradation.

Next, we estimate the lifetime of individual BrC chromophores in BBOA particles. For chamise, manzanita, and lodgepole
pine fires we measured the integrated PDA intensity over 300-700 nm for resolved BrC chromophores for up to three photolysis time points (listed in Table 3) and before photolysis. The limited number of samples and destructive nature of the chemical analysis only made it possible to do measurements for very few time points. Integrated PDA intensities as a function of irradiation time were fit assuming that the decay was exponential in time. LED lifetimes were then converted to equivalent lifetimes in the atmosphere, calculated from the average spectral flux density over June 20, 2017 in Los Angeles.
Regardless of the chromophore identities, BrC chromophores from chamise burns have shorter predicted lifetimes (0.4-0.5 days) than those from manzanita burns (0.5-0.9 days), which in turn have shorter predicted equivalent atmospheric lifetimes due to sunlight exposure than BrC from lodgepole pine burns (1.0-1.6 days), as shown in Figure 5. These lifetimes of BrC chromophores are consistent with atmospheric observations of a rapid evolution in a California wildfire, which showed that the BrC absorbance lifetime at 370 nm was 9-15 hours (Forrister et al., 2015).

The same chromophores were found to decay at different rates depending on the fuel/fire type (Figure 5). For example, very different equivalent atmospheric lifetimes due to photolysis were obtained across fuel types for veratraldehyde (#4 in Table 1, $C_9H_8O_3$), a BrC chromophore common to all three fires. One explanation is that there are multiple chromophores co-eluting at this retention time, and therefore the calculation is an average lifetime for multiple compounds. A more interesting explanation is that the surrounding matrix could affect the rate of photolysis for individual chromophores by several possible
mechanisms. First, different matrices could quench the electronic excitation in the chromophores to a different extent. Another possibility is that photodegradation of BrC chromophores could be not direct but rather occurring through condensed-phase photosensitized reactions (Malecha and Nizkorodov, 2017; Monge et al., 2012), in which case the rate of decomposition would depend on concentration of photosensitizers in the samples as well as viscosity of the material (Hinks et al., 2016; Kaur et al., 2019). Lastly, other absorbing species, such as black carbon could be shielding BrC chromophores
from irradiation, altering the amount of radiation absorbed by BrC chromophores. Given the different mechanisms, the potential contributions from each are difficult to distinguish in this study. The particle matrix is different for all three BBOA particle samples and could contribute to the very different equivalent atmospheric lifetimes of individual BrC chromophores observed in Figure 5.





We also estimated the decay lifetime for the overall BrC absorption, integrated over 300-700 nm, from different fuel types. In these experiments, BBOA filters were irradiated with a filtered xenon arc lamp, which gave a spectral flux density more similar to the sun, although more intense (Figure 1). The advantage of taking transmission spectra directly through the filters is that it makes it possible to monitor photodegradation of BrC absorption at several irradiation times, which is not possible with the solution-phase spectrophotometry, which irreversibly destroys the filter sample by extraction. The filter transmission spectra indicated that the decay of absorbance was not actually exponential. After a certain irradiation time, the BrC absorbance no longer decreased, as observed for the samples from subalpine fir and longleaf pine burns. For example, in Figure S2, after 21 hours the "baseline BrC" level has already been reached, as revealed by the next measurement at 33 hours. The absorbance decreased 70% before it reached the baseline BrC level for subalpine fir, and 60% for longleaf pine. For estimates of the BrC absorbance lifetimes, we used only the time before reaching the final light-absorbance state. Table 4 summarizes the resulting lifetimes for BrC from four fuel types, longleaf pine, juniper, lodgepole pine, and subalpine fir.

Once there was no further significant change in the transmission spectrum, the filter was extracted for the solution phase UV-Vis measurement, in order to compare the spectra obtained from the filter and in solution. The solution-phase spectra exhibited a reduction in the absorbance almost exactly to that observed in the filter transmission spectra (Figure S2). However, there were differences in the shape of the spectra – there was no measurable absorbance above 550 nm in the extracted samples, but filter samples absorbed even at these long wavelengths (Figure S2). It is likely that the extraction from the filter was not complete, and some of the absorbers remained on the filter after the extraction. The latter is another advantage of doing these experiments with filter samples as opposed to their solvent extracts.

BBOA from subalpine fir (litter + other components) had the shortest equivalent absorption lifetime at 10 days, and ponderosa pine (litter + canopy) having the next shortest equivalent absorption lifetime at 17 days. Different ecosystem biomass components were burned in the long leaf pine fire, such as duff, litter, and canopy, and had the next longest absorption lifetime of 25 days. The longest living BrC absorbance, at 41 days, was observed for the sample from juniper (canopy only) burn. Fuel components appear to affect BrC absorption lifetimes, as it does seem that non-canopy fuel components, such as litter and duff lower the BrC absorption lifetimes. However, it is difficult to correlate the BrC absorption lifetimes with quantitative measures such as NO levels or MCE (Table S1). Table S1 shows that the peak NO level was lower for long leaf pine (0.67 ppmv) compared to juniper (1.72 ppmv) and ponderosa pine (1.61 ppmv), suggesting less flaming combustion may have occurred for the long leaf pine fire (although this is not reflected in the MCE trends). Regardless, the data suggests that BrC absorption can be long-lived from direct photodegradation.

In general, the lifetimes for the loss of the absorbance integrated over 300-700 nm (Table 4) are much longer than those of individual chromophores (Figure 5). There are two likely reasons for that. First, the photodegradation of individual chromophores creates product(s) that may also absorb in the same wavelength range. The integrated BrC absorption (300-700 nm) may go to zero only after the compounds go through several stages of photodegradation, finally resulting in



products that no longer absorb above 300 nm. Second, equation 1 that we use to estimate lifetimes does not take into account photochemical quantum yields, which tend to increase greatly at shorter wavelengths. The LED, which was used in measurements of lifetimes of individual chromophores, has a higher density of higher energy photons compared to the Xe lamp (Figure 1), which could accelerate photodegradation.

The lifetimes for BrC photobleaching due to UV irradiation (10 to 41 days) are longer than what other studies have observed or approximated for other aging mechanisms. Lin et al. 2016 found that peat and ponderosa pine BBOA had similar half-lives of around 16 hours based on absorption coefficients at 300 nm. However, in Lin et al. 2016 BBOA was extracted and irradiated in solution where photodegradation could occur more rapidly (Lignell et al., 2014). Forrister et al. 2015 collected filter samples in the plumes of wildfires with different transport times during the SEAC4RS campaign, and found that the
BrC absorbance lifetime at 370 nm was 9-15 hours. Similarly, Selimovic et al. 2019 found a significant decrease in AAE after 10 hours of daytime aging during a wildfire event in the Northwestern US. Sumlin et al., 2017 aged smoldering peat BBOA in an OFR, and reported a decrease of ~40-50% in the aerosol mass absorption coefficients at 375 nm and 405 nm after 4.5 equivalent aging days. They attributed this decrease to fragmentation of BrC chromophores due to photooxidation (oxidation by gaseous OH). Based on the comparison of these observations, photooxidation could be a more important aging
mechanism affecting BrC absorption lifetimes than the UV-induced photochemical processes inside the particles.

## 4 Conclusions and Implications

BBOA particles from laboratory burns of twelve forest fire fuels collected around the United States were analyzed for BrC chromophores. Biomass fuels spanned plant types (gymnosperm versus angiosperm) and ecosystem components (duff, litter,
canopy, etc.). BrC chromophores were grouped among classes, including: lignin pyrolysis products, lignin-derived, distillation (coumarins and flavonoids), nitroaromatics, and PAHs. While most BrC chromophore classes were observed in all burns, regardless of fuel type, there were specific BrC chromophores that were divided across angiosperm (flowering) and gymnosperm (conifer) lines. For example, sinapaldehyde was mainly observed in BBOA particles when angiosperm fuels were burned and coniferaldehyde when gymnosperm fuels were burned. Additionally, there were flavonoids specific to
conifers, tentatively kaempferol and diosmetin (Table 1, chromophores 11 & 14), and unique to angiosperms such as chromophore 16. PAHs are largely angiosperm BrC chromophores, showing up mainly for sagebrush, chamise, and ceanothus fuels. There are some BrC chromophores that are only appreciably observed in a single fuel type/burn; many of these are likely distillation or lignin-derived products. The most absorbing of these BrC chromophores are components of the angiosperm BBOA particles (Table 2).

UV irradiation of BBOA particles from different fuels directly on filters removes some BrC chromophores but some appear to be photo-stable, specifically, lignin-derived compounds (including lignin-pyrolysis products) and flavonoids.



Interestingly, individual BrC chromophore lifetimes varied based on the fuel burned and perhaps the underlying combustion conditions, rather than just the structure of the chromophore. Part of the reason is that co-elution of chromophores with different stabilities complicates measurements of individual chromophore lifetimes. In addition, indirect photolysis mechanisms, such as photosensitized reactions, the release of absorbed energy by a neighboring molecule as heat depending on intermolecular forces, and shielding of light by other absorbing molecules could change depending on the specific BBOA material. The BrC chromophores of chaparral fuels had shorter equivalent photochemical lifetimes compared to BBOA generated from the canopies of conifer fuel types. On the whole, these results suggest that some of the primary BrC chromophores may be destroyed by UV irradiation after several hours.

Despite the rapid change in the absorbance of individual chromophores, the overall integrated BrC absorbance from 300 nm to 700 nm decayed with a much longer lifetime of 10 days to 41 days. These observations contrast with individual chromophores in particles that decayed on the time scale of 0.4 to 1.6 days. Taken together the two types of photolysis experiments suggest that the absorption by the complete pool of BrC compounds persists during irradiation longer than the individual BrC chromophores detected. Our findings also show that ecosystem components, and the combustion conditions they create, could influence the apparent BrC absorption lifetimes. BrC from the subalpine fir mix burned with more smoldering combustion and had the shortest equivalent lifetime of 10 days, while BBOA from the juniper and Lodgepole canopy fuels had longer BrC absorption lifetimes of 25-41 days. The canopy fuels contributed to more flaming combustion. These fairly long BrC absorption lifetimes suggest that the BrC was likely photostable upon direct photolysis, and other chemical aging mechanisms such as OH oxidation may be more important under atmospheric conditions. Based on these results, modelers should first focus on chemical aging mechanisms other than photolysis, such as heterogeneous oxidation by OH.

**Author contributions**

LF, PL and AL collected and analyzed particulate matter samples, and JR, VS and RY analyzed gaseous composition of BBOA. JL, AL and SN assisted with interpretation of mass spectrometry data. LF did the photochemistry experiments and wrote the paper. All co-authors provided edits and critical feedback for the paper.

**Acknowledgements**

LTF and SAN were supported by NOAA grant NA16OAR4310102. PL, JL, and AL were supported by NOAA grant NA16OAR4310101. VS and RY were supported by NOAA-CPO grant NA16OAR4310100. We thank the USFS Missoula Fire Sciences Laboratory for their help in conducting these experiments. This work was also supported by NOAA's Climate Research and Health of the Atmosphere Initiative. The HRMS measurements were performed at the W.R. Wiley Environmental Molecular Sciences Laboratory (EMSL) – a national scientific user facility located at PNNL, and sponsored





by the Office of Biological and Environmental Research of the U.S. DOE. PNNL is operated for U.S. DOE by Battelle Memorial Institute under Contract No. DE-AC06-76RL0 1830.

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




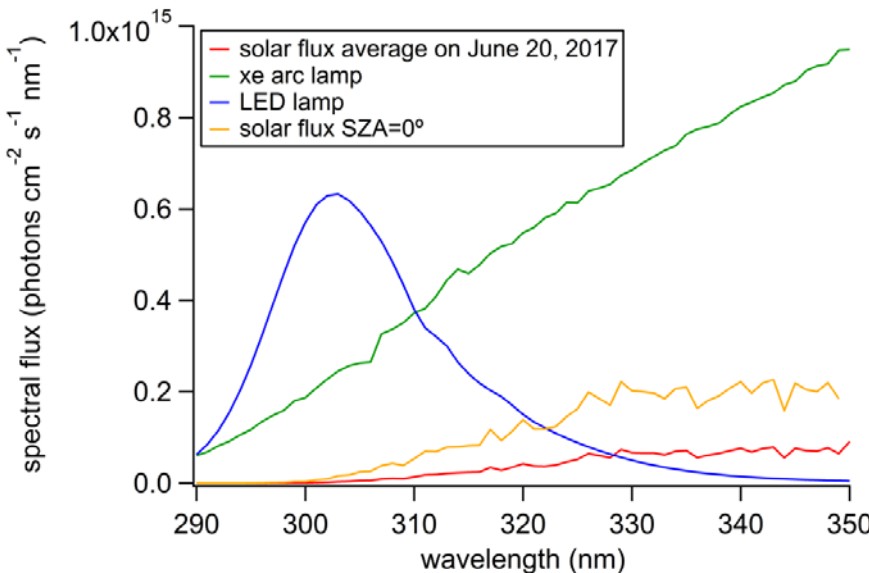

**Figure 1. Spectral flux density (photons cm$^{-2}$ s$^{-1}$ nm$^{-1}$) approximated for a solar zenith angle of 0° (orange) as well as the 24-hour average for the latitude and longitude of Los Angeles (34° latitude, 118° longitude) on June 20, 2017 (red). The spectral flux density for the 300 nm LED (blue) and the filtered Xe arc lamp (green) are also shown.**

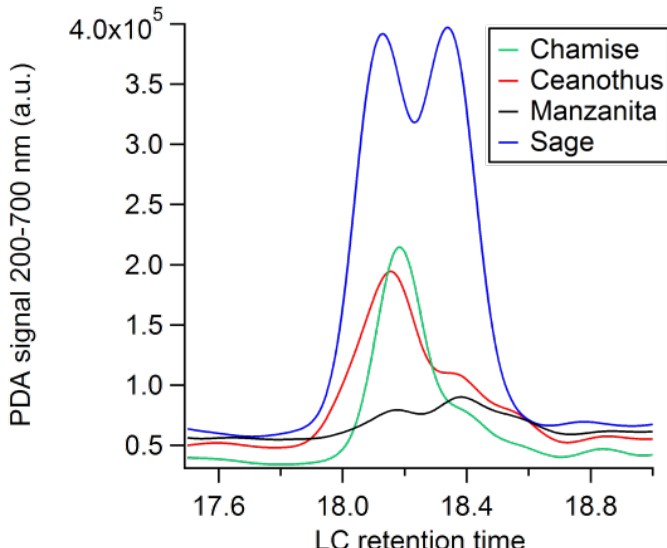

**Figure 2. Lignin pyrolysis products sinapaldehyde (C$_{11}$H$_{12}$O$_4$) and coniferaldehyde (C$_{10}$H$_{10}$O$_3$) elute at slightly different retention times, at roughly 18.1 and 18.4 min, respectively.**



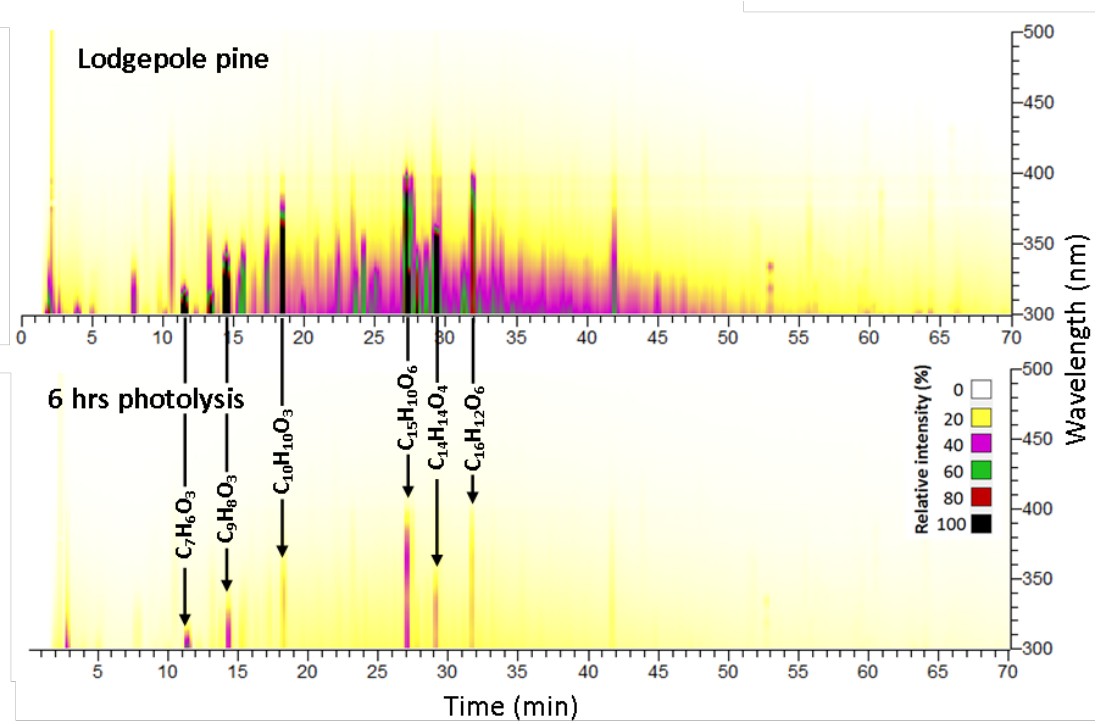

**Figure 3. BrC chromophores present in the BBOA sample before (top panel) and after (bottom panel) photolysis for a conifer fuel, lodgepole pine.**



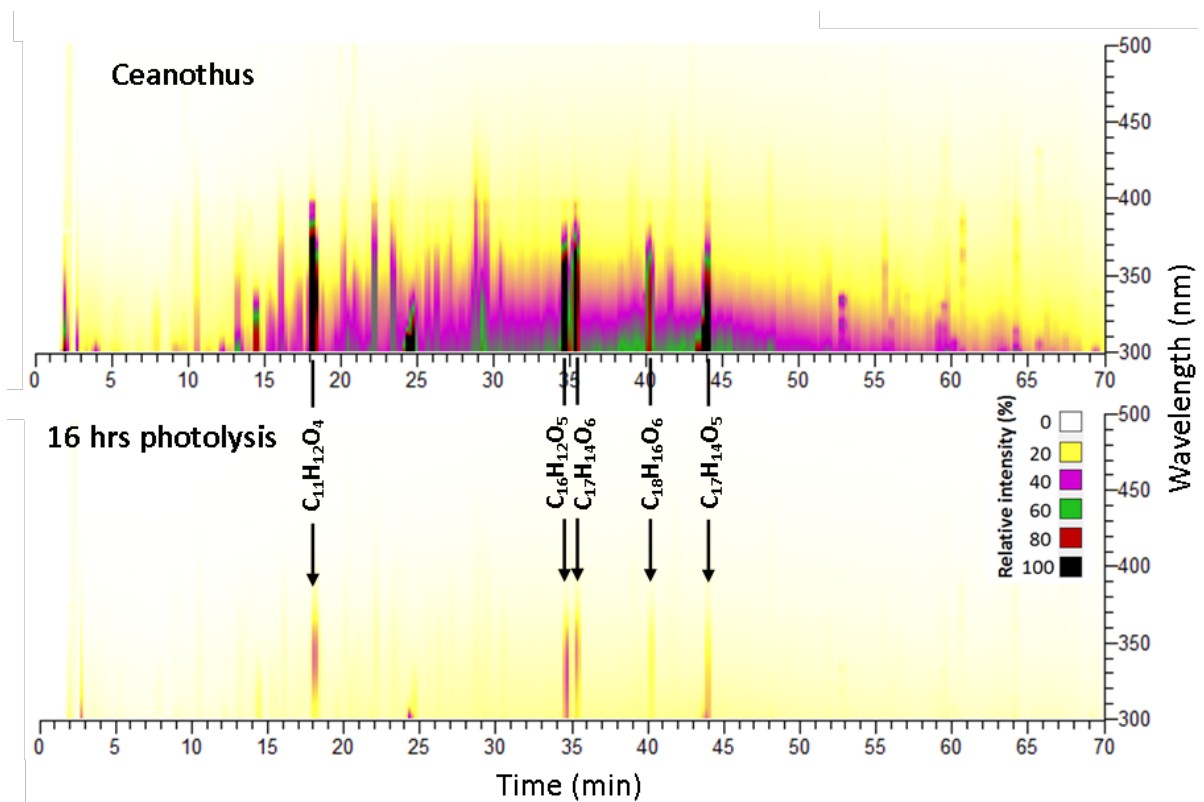

**Figure 4. BrC chromophores present in the BBOA sample before (top panel) and after (bottom panel) photolysis for angiosperm fuel, ceanothus.**

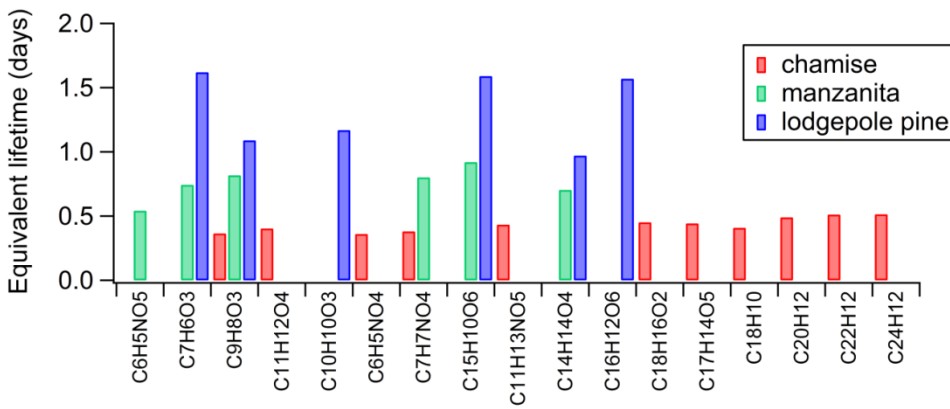

**Figure 5. Approximate atmospheric lifetimes for select individual BrC chromophores due to direct photolysis that exist in BBOA particles from chamise, manzanita, and lodgepole pine fires (the photolysis times are listed in table 3). These lifetimes are shorter than those calculated for overall BrC absorption.**



**Table 1. Chromophores common among multiple fuel types are listed by their HPLC retention times, absorption spectra, assigned elemental formulas, and examples of possible structures. The absorbance by each chromophore is binned by photodiode array absorbance normalized to the highest absorbance in each chromatogram: M-Major (75%-100%), I-Intermediate (25%-75%), or W-Weak (5%-25%). The absorption spectra of the standard may not fully match the absorption spectra of the eluents because the separation is not complete, and more than one compound may elute at any given time. The shown absorption spectra are baseline corrected by subtracting the spectrum at a nearby retention time where the PDA absorbance is low.**

| Peak # | LC RT (min) | Absorption spectrum | Elemental formula of neutral species | Examples of possible structures | subalpine fir duff | ceanothus | chamise | longleaf pine | juniper | ponderosa pine log | manzanita | lodgepole pine | engelmann spruce duff | ponderosa pine litter | douglas fir | sagebrush |
|---|---|---|---|---|---|---|---|---|---|---|---|---|---|---|---|---|
| 1 | 10.07-10.29 | | $C_8H_8O_4$ | vanillic acid | W | | | | | | W | | | W | | |
| 2 | 10.52-10.71* | | $C_6H_5NO_5$ | nitropyrogallol | | W | | I | W | | M | W | | | I | W |
| 3 | 11.37-11.61 | | $C_7H_6O_3$ | salicylic acid | | | | W | | | W | I | | | W | W |
| 4 | 14.37-14.56 | | $C_9H_8O_3$ | veratraldehyde | W | W | W | I | W | | I | M | W | W | I | W |



| Peak # | LC RT (min) | Absorption spectrum | Elemental formula of neutral species | Examples of possible structures | subalpine fir duff | ceanothus | chamise | longleaf pine | juniper | ponderosa pine log | manzanita | lodgepole pine | engelmann spruce duff | ponderosa pine litter | douglas fir | sagebrush |
|---|---|---|---|---|---|---|---|---|---|---|---|---|---|---|---|---|
| 5 | 15.17-15.46 | | $C_9H_6O_3$ | umbelliferone | W | W | | W | I | W | W | W | W | | W | W |
| 6 | 16.12-16.24 | | $C_6H_5NO_4$ | nitrocatechol | | W | I | W | | | | | | W | | I |
| 7 | 17.20-17.42 | | $C_7H_7NO_5$ | hydroxynitroguaiacol | W | | | W | W | I | W | W | W | W | W | W |
| 8 | 18.13-18.18 | | $C_{11}H_{12}O_4$ | sinapaldehyde | | M | M | W | | M | W | | M | | | M |





| Peak # | LC RT (min) | Absorption spectrum | Elemental formula of neutral species | Examples of possible structures | subalpine fir duff | ceanothus | chamise | longleaf pine | juniper | ponderosa pine log | manzanita | lodgepole pine | engelmann spruce duff | ponderosa pine litter | douglas fir | sagebrush |
|---|---|---|---|---|---|---|---|---|---|---|---|---|---|---|---|---|
| 9 | 18.33-18.47 |  | $C_{10}H_{10}O_3$ | coniferaldehyde | M | I | W | M | M | M | W | M | | M | M | M |
| 10 | 23.32-23.49 | | $C_7H_7NO_4$ | methyl nitrocatechol | | W | W | W | | W | W | W | | W | | I |
| 11 | 27.15-27.29 | | $C_{15}H_{10}O_6$ | kaempferol | | | | I | W | | W | M | | W | I | |
| 12 | 28.80-28.85 | | $C_{11}H_{13}NO_5$ | | | W | M | | | | | | | | | M |





| Peak # | LC RT (min) | Absorption spectrum | Elemental formula of neutral species | Examples of possible structures | subalpine fir duff | ceanothus | chamise | longleaf pine | juniper | ponderosa pine log | manzanita | lodgepole pine | engelmann spruce duff | ponderosa pine litter | douglas fir | sagebrush |
|---|---|---|---|---|---|---|---|---|---|---|---|---|---|---|---|---|
| 13 | 29.08-29.34 | | $C_{14}H_{14}O_4$ | nodakenetin | M | | | M | M | I | W | M | I | M | M | I |
| 14 | 31.85-31.90 | | $C_{16}H_{12}O_6$ | diosmetin | | | | W | | | | I | | W | | |
| 15 | 41.63-41.98 | | $C_{18}H_{16}O_2$ $C_{17}H_{10}O$ $C_{16}H_{10}O$ | | | W | W | W | | W | | W | W | W | W | W |
| 16 | 43.95-44.05 | | $C_{17}H_{14}O_5$ $C_{19}H_{10}O$ | | | I | I | | | | | | | | | W |
| 17 | 55.51-55.63 | | $C_{18}H_{10}$ | | | W | I | | W | | | | | | | I |


| Peak # | LC RT (min) | Absorption spectrum | Elemental formula of neutral species | Examples of possible structures | subalpine fir duff | ceanothus | chamise | longleaf pine | juniper | ponderosa pine log | manzanita | lodgepole pine | engelmann spruce duff | ponderosa pine litter | douglas fir | sagebrush |
|---|---|---|---|---|---|---|---|---|---|---|---|---|---|---|---|---|
| 18 | 59.54-59.63 | | $C_{20}H_{12}$ $C_{20}H_{14}$ $C_{20}H_{10}$ | | | W | W | | | | | | | | | W |
| 19 | 60.61-60.78 | | $C_{16}H_{12}$ | | | W | W | | | | | | | | | W |
| 20 | 63.15-63.23 | | $C_{25}H_{12}O$ $C_{23}H_{10}O_2$ | | | | W | | | | | | | | | W |
| 21 | 64.11-64.28 | | $C_{22}H_{12}$ | | | W | I | | | | | | | | | I |



| Peak # | LC RT (min) | Absorption spectrum | Elemental formula of neutral species | Examples of possible structures | subalpine fir duff | ceanothus | chamise | longleaf pine | juniper | ponderosa pine log | manzanita | lodgepole pine | engelmann spruce duff | ponderosa pine litter | douglas fir | sagebrush |
|---|---|---|---|---|---|---|---|---|---|---|---|---|---|---|---|---|
| 22 | 65.61-65.74* | | $C_{24}H_{14}$ $C_{23}H_{14}$ $C_{27}H_{12}O$ | | | | W | | | | | | | | | W |
| 23 | 67.37-67.46* | | $C_{24}H_{12}$ $C_{24}H_{14}$ $C_{23}H_{12}$ | | | | W | | | | | | | | | W |
| 24 | 68.15-68.27 | | $C_{23}H_{14}$ | | | | W | | | | | | | | | W |
| 25 | 71.77-71.89* | | $C_{26}H_{14}$ $C_{26}H_{12}$ | | | | W | | | | | | | | | W |



| Peak # | LC RT (min) | Absorption spectrum | Elemental formula of neutral species | Examples of possible structures | subalpine fir duff | ceanothus | chamise | longleaf pine | juniper | ponderosa pine log | manzanita | lodgepole pine | engelmann spruce duff | ponderosa pine litter | douglas fir | sagebrush |
|---|---|---|---|---|---|---|---|---|---|---|---|---|---|---|---|---|
|  |  |  |  | <br><br> |  |  |  |  |  |  |  |  |  |  |  |  |





Table 2. Chromophores found appreciably in only one fuel type listed by their HPLC retention time, absorption spectra, assigned elemental formulas, and examples of possible structures. The absorbance by each chromophore is binned by photodiode array absorbance normalized to the highest absorbance in each chromatogram: M-Major (75%-100%), I-Intermediate (25%-75%), or W-Weak (5%-25%).

| Peak # | LC RT (min) | Absorption spectrum | Elemental formula | Intensity bin | Examples of possible structures | Fuel type |
|---|---|---|---|---|---|---|
| 26 | 14.22 | | $C_{10}H_8O_4$ | I | scopoletin | sagebrush |
| 27 | 17.08 | | $C_{13}H_{14}O_4$ | W | | chamise |
| 28 | 18.87 | | $C_{10}H_{10}O_2$ | W | | subalpine fir duff |
| 29 | 21.82 | | $C_7H_7NO_5$ (isomer of methyl nitrocatechol) | W | | sagebrush |



| Peak # | LC RT (min) | Absorption spectrum | Elemental formula | Intensity bin | Examples of possible structures | Fuel type |
|---|---|---|---|---|---|---|
| 30 | 22.21 | | $C_8H_9NO_5$ | W | Dimethoxynitrophenol | chamise |
| 31 | 22.43 | | $C_{12}H_{12}O_4$ | W | | lodgepole pine |
| 32 | 23.65 | | $C_{18}H_{18}O_5$ | W | | lodgepole pine |
| 33 | 24.42 | | $C_{10}H_{10}O_5$ | W | | ceanothus |



| Peak # | LC RT (min) | Absorption spectrum | Elemental formula | Intensity bin | Examples of possible structures | Fuel type |
|---|---|---|---|---|---|---|
| 34 | 26.17 | | $C_{20}H_{22}O_6$ | W | | ponderosa pine log |
| 35 | 26.95 | | $C_{11}H_6O_3$ $C_{17}H_{14}O_8$ | W | | lodgepole pine |
| 36 | 30.35 | | $C_{12}H_{13}NO_4$ $C_{16}H_{16}O_6$ | W | | ponderosa pine log |
| 37 | 32.55 | | $C_{10}H_7NO_3$ | W | | sagebrush |





| Peak # | LC RT (min) | Absorption spectrum | Elemental formula | Intensity bin | Examples of possible structures | Fuel type |
|---|---|---|---|---|---|---|
| 38 | 34.69 | | $C_{16}H_{12}O_5$ | W | | ceanothus |
| 39 | 35.25 | | $C_{11}H_9NO_3$ | W | | chamise |
| 40 | 35.37 | | $C_{17}H_{14}O_6$ | I | | ceanothus |
| 41 | 39.38 | | $C_{16}H_{10}O_3$ | W | | chamise |





| Peak # | LC RT (min) | Absorption spectrum | Elemental formula | Intensity bin | Examples of possible structures | Fuel type |
|---|---|---|---|---|---|---|
| 42 | 40.22 | | $C_{18}H_{16}O_6$ | I | | ceanothus |
| 43 | 40.53 | | $C_{17}H_{20}O_4$ | W | | ponderosa pine log |
| | 44.26 | | $C_{32}H_{28}O_4$ | W | | subalpine fir duff |
| | 45.7 | | $C_{22}H_{26}N_4O$ | W | | subalpine fir duff |



| Peak # | LC RT (min) | Absorption spectrum | Elemental formula | Intensity bin | Examples of possible structures | Fuel type |
|---|---|---|---|---|---|---|
| | 49.05 |  | $C_{20}H_{24}O_4$ | W | | ponderosa pine log |

*Lin et al. (2018)





**Table 3. Filter irradiation times for different samples used to estimate lifetimes of individual chromophores.**

| Fuel type | Irradiation time #1 (hrs) | Irradiation time #2 (hrs) | Irradiation time #3 (hrs) | Figure used |
|---|---|---|---|---|
| Lodgepole pine | 6 | 16.8 | | 3,5 |
| Ceanothus | 16 | | | 4 |
| Chamise | 1 | 3 | 12 | 5 |
| Manzanita | 6 | 16.8 | | 5 |

**Table 4. Lifetimes for the loss of the measured integrated absorbance from 300 to 700 nm. The results are expressed in equivalent days of solar exposure to either time-averaged solar flux in Los Angeles (middle column) or peak solar flux at SZA=0° (right column). The lifetimes were calculated from the transmission spectra measured for particles on PTFE filters. The irradiation was done in the condensed phase, on the filter, for all samples.**

| Fuel type | BrC lifetime based on condensed phase measurements Average LA (equivalent days) | BrC lifetime based on condensed phase measurements SZA=0° (equivalent days) |
|---|---|---|
| Longleaf pine | 25 | 8.5 |
| Juniper | 41 | 14 |
| Ponderosa pine litter | 17 | 6.0 |
| Subalpine fir duff | 10 | 3.4 |