# Peer review of "Molecular composition and photochemical lifetimes of brown carbon chromophores in biomass burning organic aerosol"

_Atmospheric Chemistry and Physics, 2019_

## Referee Comment (RC1) · Anonymous Referee #1 · 18 Jul 2019

Fleming et al. present brown carbon observations from biomass burning emissions collected during FIREX. Smoke samples were analyzed using HPLC/PDA/HRMS to determine which compounds absorb in the 300-700 nm wavelength and their corresponding molecular formula. 46 brown carbon chromophores were separated and roughly identified. These compounds were further divided into classes such as lignin pyrolysis products and nitroaromatics. In addition, filters were photochemically aged. Two techniques were used to age the filters (1) LED light exposure and liquid extraction for HPLC/PDA/HRTMS analysis (2) irradiation of filters with xenon arc lamp and analysis by an absorption spectrometer. Their results indicate that smoke produced from specific fuels or fuel mixtures photochemically age at different rates, between 10-

41 days. Canopy fuels showed longer photochemical lifetimes than subalpine fir fuel mixtures. The authors point out that chemical aging mechanisms likely play a larger role in aging on these time scales than photochemical processes.

Overall, this study provides very interesting information on specific brown carbon compounds found in fresh and aged emissions. The topic of the paper is appropriate for ACP and is clearly written. The authors should address the below comments.

Major comments:

With regards to calculating aging times: The authors integrated over the spectral flux for the two types of lights and normalized this to the spectral flux of the sun. However, shining light onto a particle collected onto a filter is different than shining a light on an aerosol particle. The filter will absorb and reflect light back through the collected particle, which will alter the amount of radiation the particles experiences compared to aerosol particles—up to double the amount of radiation. Do the authors know how the filter will alter calculated photochemical lifetimes?

Page 12, line 366: The authors comment how the same chromophore exhibited different lifetimes between fuels. Judging from figure 5, the range of lifetimes is within a factor of 2-3 (0.4-1.2 days). The authors provide several reasonable explanations on why the wide range of lifetimes for the same chromophore (or compounds that elute at the same time). However, the authors in the previous paragraph note that brown carbon chromophores from chamise burns have shorter lifetimes than lodgepole pine burns. Can the authors comment on the uncertainty range for these lifetimes as it seems like the variation for just one chromophore is enough to explain and differences in specific chromophore lifetimes observed between fuels.

Minor comments: Page 2, line 47: "Wildfires continue to..." This sentence semi-repeats previously stated information.

Page 2, line 59: "large effect on radiative forcing" In which direction?

[Figure]

Page 2, line 65: "broader range" broader range than what?

Page 4, line 123: "irradiated BBOA water extracts..." The phrasing of these sentence is a bit confusing as it seems like the solution just randomly lost WSOC.

Page 4, line 126: "produced from Alaskan peat" produced from burning Alaskan peat

Page 5, line 138: the semi colon is not used properly

Page 6, line 173: How long were the filters stored for before analysis? If it was a long time (months later), do the authors know/guess how these chromophores degraded with time?

Page 9, line 252: For the structures of compounds given in Table 1, did the authors run standards to try to better confirm the identities/structures of the reported compounds? By reference spectrum, do the authors mean spectra from running standard compounds?

Page 14 line 418: Please provide another sentence or phrase explaining why photodegradation occurs more rapidly in solution.

Page 14 line 419: no comma after campaign

Page 14, line 420: What does AAE stand for?

Figure 1 and 2: Since there are not too many lines, please make the lines distinguishable for black and white printers.

Figure 2: Maybe label on the graph the two compounds.

Figure 3 and 4: There appears to be a bracket-shaped line border around these figures. Can this be removed? Also, caption says photolysis but which type of photolysis?

Figure 5: Along with the comment from before, it would be useful to have uncertainty bars on this figure so the reader knows what constitutes a pattern or not.

Table 1: Why do the peak # stop after 43?

Table 3: What does "Figured used" column mean? Is this important information?

Table 4: The caption says irradiation was done for condensed phase, filter, and all samples. This is confusing as the columns only reference condensed phase. Overall, the column headings are confusing. Should average not be capitalized? And averaged?

Supporting Information:

Figure S1: Please make this graph more readable. Which equation goes to which line? Which lines describes which points? There appears to be only two data points on the condensed phase line (not counting 0,0). Why were more points not taken? Seems like 0,0 is dominating the trend line which makes the measured points seem very off. Also please add in the text that this method was used for both types of aging experiments (specific chromophores and overall BrC) unless this is not the case.

Relating to the previous comment, there seems to be pretty high uncertainty in estimating lifetimes as this Figure S1 shows 1-2 data points and a linear regression. Please add a significant discussion on uncertainty bounds.

Is condensed phase filter-based? In the main text (page8 line 239), you refer to both filter-based and condensed phase. Could you please clarify which term refers to what?
* * *

---

## Referee Comment (RC2) · Anonymous Referee #2 · 23 Jul 2019

This manuscript by Fleming et al. reports an investigation of the chemical nature of brown carbon (BrC) emitted from the burning of biomass fuels, including coniferous and flowering plants, as well as different ecosystem components including duff, litter, and canopy. Using a combination of analytical techniques, BrC chromophores are categorized into several classes: 1) lignin-derived, 2) distillation products, 3) nitroaromatics, and 4) polycyclic aromatic hydrocarbons. An extensive list of chromophores with their relative contribution to overall BrC absorption is presented. The study also investigates the evolution of the BrC chromophores due to photolysis-induced photobleaching. The major findings are that 1) the BrC absorption lifetimes are variable and dependent on the type of fuel burned and 2) other chemical aging mechanisms in the atmosphere,

such as OH oxidation, are more important in the evolution of BrC chromophores than photolysis. This paper provides substantial contribution to the understanding of the chemical nature and lifetime of BrC in biomass burning emissions. I encourage the authors to address the following comments:

1) The extraction procedure in the first paragraph of section 2.2 requires further elaboration. What is the rationale behind using the 3 organic solvents in these proportions? Is that based on an already established protocol? Have you verified that none of the BrC precipitates out of the water+DMSO mixture after evaporating the organic solvents? Why are there different extraction protocols for the fresh (water+DMSO) and photolyzed (only DMSO) particles?

2) Line 235: Along the same lines, why is the extraction procedure for UV-vis different from HPLC/PDA/HRMS?

3) The binning of the chromophores into Major, Intermediate, and Weak is informative, but can be confusing. I think it is helpful to stress that these categories involve the combined effects of abundance and absorption efficiency. For instance, an abundant chromophore with moderate absorption efficiency can exist in the same bin as a less abundant chromophore with high absorption efficiency.

4) In addition to integrating the DPA signal over 300nm – 700 nm, can the authors also report wavelength-dependent information? Are there significant differences in AAE across the different chromophores? If yes, integrating absorption over say 4 sections (UV, short visible, mid visible, long visible) can provide interesting information. For example, a chromophore under category W but with say a relatively large AAE might be an I or an M in the UV range.
* * *

---

## Referee Comment (RC3) · Anonymous Referee #3 · 24 Jul 2019

The work of Fleming et al. describes a series of measurements made on filters collected from laboratory burning. They use chemical separation followed by UV-vis absorption spectrophotometry and high-resolution mass spectrometry to identify and classify brown carbon (BrC) chromophores from different fuels. They also age the filters using two different lamps, directly measuring transmission and looking at extracts. They observe the photodegradation of BrC chromophores, leading to a persistent fraction of BrC. I recommend the changes detailed below be made to the manuscript prior to consideration for publication in ACP.

General comments

The Authors noted a decrease in absorption following photolysis that leveled off. This observation of a longer-lived fraction of BrC is consistent with several measurements in the lab (e.g. Wong et al. (2017)) and real environment (e.g. Forrister et al. (2015) and Di Lorenzo et al. (2017)). The previous work is not effectively reflected in the text of the paper. For example, lines 60-62 suggest that field measurements demonstrate a short lifetime for BrC. While observations of a short-lived fraction have been made, long-lived fractions have also been observed. In addition, the results described leading up to line 406 should be discussed in the context of this literature.

More information should be provided about the solvents chosen for filter extraction. On line 139, the Authors say, "filters were extracted by solvents with a range of polarities". Only two solvent mixtures were used, and these were used to extract different samples. A mixture of dichloromethane/acetonitrile/hexanes was used to extract samples prior to HPLC/PDA/HRMS analysis, while a mixture of methanol/acetonitrile/hexane was used to extract PTFE filters following transmission measurements. A justification should be provided for the choice of each solvent mixture. The impact of the solvent choice on the results described in lines 393-395 should also be discussed.

Specific comments

Line 70: Typo in polycyclic aromatic hydrocarbons.

Line 122-123: Wong et al. showed not just a decrease in WSOC, but also a decrease in BrC absorption.

Line 266: Should be "one fewer methoxy ring substituent".

Line 318: Saying that PAHs are "stable during atmospheric transport" is an over-simplification. Although they may end up in pristine regions, the conditions that allow them to undergo long range transport are complex. The review from Keyte et al. (2013) and recent work from Zhou et al. (2019), along with other relevant work, should be consulted and discussed in this section.

Figure 1: Suggest this figure could be relocated to the SI.

Sources Cited

Forrister, H., Liu, J., Scheuer, E., Dibb, J., Ziemba, L., Thornhill, K. L., Anderson, B., Diskin, G., Perring, A. E., Schwarz, J. P., Campuzano-Jost, P., Day, D. A., Palm, B. B., Jimenez, J. L., Nenes, A. and Weber, R. J.: Evolution of brown carbon in wildfire plumes, Geophys Res Lett, 42(11), 4623–4630, doi:10.1002/2015GL063897, 2015.

Keyte, I. J., Harrison, R. M. and Lammel, G.: Chemical reactivity and long-range transport potential of polycyclic aromatic hydrocarbons-a review, Chem Soc Rev, 42(24), 9333–9391, doi:10.1039/c3cs60147a, 2013.

Di Lorenzo, R. A., Washenfelder, R. A., Attwood, A. R., Guo, H., Xu, L., Ng, N. L., Weber, R. J., Baumann, K., Edgerton, E. and Young, C. J.: Molecular size separated brown carbon absorption for fresh and aged biomass burning plumes at multiple field sites, Environ Sci Technol, 51, 3128–3137, doi:10.1021/acs.est.6b06160, 2017.

Wong, J. P. S., Nenes, A. and Weber, R. J.: Changes in light absorptivity of molecular weight separated brown carbon due to photolytic aging, Environ Sci Technol, 51(15), 8414–8421, doi:10.1021/acs.est.7b01739, 2017.

Zhou, S., Hwang, B. C. H., Lakey, P. S. J., Zuend, A., Abbatt, J. P. D. and Shiraiwa, M.: Multiphase reactivity of polycyclic aromatic hydrocarbons is driven by phase separation and diffusion limitations, Proc Natl Acad Sci, 116, 11658–11663, doi:10.1073/pnas.1902517116, 2019.

---

## Author Comment (AC1) · 2 Nov 2019

**Reviewer #1**

We would like to thank the Reviewers for their constructive feedback. The reviewer comments are reproduced below in black, and our responses are provided below each comment in blue. We will also be uploading a track-changed version of the manuscript along with all responses as required by ACP.

Major comments:

With regards to calculating aging times: The authors integrated over the spectral flux for the two types of lights and normalized this to the spectral flux of the sun. However, shining light onto a particle collected onto a filter is different than shining a light on an aerosol particle. The filter will absorb and reflect light back through the collected particle, which will alter the amount of radiation the particles experiences compared to aerosol particles up to double the amount of radiation. Do the authors know how the filter will alter calculated photochemical lifetimes?

This is an excellent point, which we neglected to explain in this paper. Presser et al. (2014) measures the absorption enhancement due to scattering by different kinds of filters coated with nigrosine dye. They found that the absorption enhancement was highest for the Teflon filters, and was highly dependent upon the mass loading and temperature, and perhaps even pore size. This is an additional source of error in our lifetimes since we are not able to quantify the absorption enhancement specifically for our filters. We now clarify in the paper that these lifetimes of BrC absorption and chromophores are lower limits. It is now mentioned three times in the paper, once in the Experimental section and twice in the Results and Discussion section (Pages 9, 12, 13).

Page 12, line 366: The authors comment how the same chromophore exhibited different lifetimes between fuels. Judging from figure 5, the range of lifetimes is within a factor of 2-3 (0.4-1.2 days). The authors provide several reasonable explanations on why the wide range of lifetimes for the same chromophore (or compounds that elute at the same time). However, the authors in the previous paragraph note that brown carbon chromophores from chamise burns have shorter lifetimes than lodgepole pine burns. Can the authors comment on the uncertainty range for these lifetimes as it seems like the variation for just one chromophore is enough to explain and differences in specific chromophore lifetimes observed between fuels.

This comment must have been triggered by our lack of error bars in the initial submission. We have added error bars for the lifetimes provided in the paper (Figure 5 and Table 4). Error bars were calculated from the standard error of the linear trendline's slope, which is the first order rate constant. This error only accounts for the uncertainty in describing the measurements with the model. There are uncertainties in the measurement method that cannot necessarily be quantified, such as differences in extraction efficiency between photolyzed and unphotolyzed parts of the same filter and the final concentration volume of HPLC/PDA/HRMS samples.

Minor comments:

Page 2, line 47: "Wildfires continue to. . ." This sentence semi-repeats previously stated information.

We agree that the sentence is somewhat repetitive and disrupts the flow of the paragraph. It was deleted.

Page 2, line 59: "large effect on radiative forcing" In which direction?

It is now clarified in the paper with "positive."

Page 2, line 65: "broader range" broader range than what?

We mean broad range in terms of the diversity of molecules and combustion sources which can produce different molecules. We hope it is clearer now.

Page 4, line 123: "irradiated BBOA water extracts. . ." The phrasing of these sentences a bit confusing as it seems like the solution just randomly lost WSOC.

Thank you, we clarified that this is in response to irradiation of the extract.

Page 4, line 126: "produced from Alaskan peat" produced from burning Alaskan peat

Clarified.

Page 5, line 138: the semi colon is not used properly

We replaced the semicolon with a comma in the revised sentence.

Page 6, line 173: How long were the filters stored for before analysis? If it was a long time (months later), do the authors know/guess how these chromophores degraded with time?

In the revised manuscript, we added that the BrC chromophore analysis was done no more than 2 months later. Since the samples were always frozen, we do not expect BrC chromophores to decay on this timescale (although no one has ever verified the chemistry occurring on frozen filters!). The partitioning of semi-volatile components may be different from the fresh emissions, but we are not as concerned with comprehensive particle composition in this paper.

Page 9, line 252: For the structures of compounds given in Table 1, did the authors run standards to try to better confirm the identities/structures of the reported compounds? By reference spectrum, do the authors mean spectra from running standard compounds?

"Reference" was corrected to "standard." Standards were run for select compounds based on availability. The caption in Table 1 was clarified to show that the standard spectrum is in blue.

Page 14 line 418: Please provide another sentence or phrase explaining why photodegradation occurs more rapidly in solution.

The sentence has been added to and now reads:

"However, in Lin et al. (2016) BBOA was extracted and irradiated in solution where photodegradation could occur more rapidly due to molecular diffusion (Lignell et al., 2014)."

Page 14 line 419: no comma after campaign

Done.

Page 14, line 420: What does AAE stand for?

Thanks. This was corrected.

Figure 1 and 2: Since there are not too many lines, please make the lines distinguishable for black and white printers.

Thanks for the suggestion. With four traces, the authors prefer colors rather than using dash/grayscale color variations.

Figure 2: Maybe label on the graph the two compounds.

Thanks, we took your suggestion.

Figure 3 and 4: There appears to be a bracket-shaped line border around these figures. Can this be removed? Also, caption says photolysis but which type of photolysis?

These lines seem to appear in PDF conversion. We will work with the publisher to edit them out from the final version. Thanks, we have clarified that this is 300 nm photolysis.

Figure 5: Along with the comment from before, it would be useful to have uncertainty bars on this figure so the reader knows what constitutes a pattern or not.

Please see the above comment. In short, we did add uncertainty bars.

Table 1: Why do the peak # stop after 43?

Thanks, this was corrected.

Table 3: What does "Figured used" column mean? Is this important information?

The title of the column is now "Sample used in figure #" and it was moved to the SI (now Table S2).

Table 4: The caption says irradiation was done for condensed phase, filter, and all samples. This is confusing as the columns only reference condensed phase. Overall, the column headings are confusing. Should average not be capitalized? And averaged?

The column headings were simplified/changed to "BrC absorption lifetime averaged LA (equivalent days)."

Supporting Information:

Figure S1: Please make this graph more readable. Which equation goes to which line? Which lines describes which points? There appears to be only two data points on the condensed phase line (not counting 0,0). Why were more points not taken? Seems like 0,0 is dominating the trend line which makes the measured points seem very off. Also please add in the text that this method was used for both types of aging experiments (specific chromophores and overall BrC) unless this is not the case.

The trend lines are very similar, and we agree it is difficult to tell which is which, although the $R^2$ is a good indicator that it's the solution phase measurement (since there's only two points). We agree that more points should have been taken to be more confident in the trend. However, it is important that the trend line go through 0,0 and it is important to keep in mind that the photolysis will likely not follow a first order decay process perfectly, and some variation is expected.

The following is said in the text (SI p3). We have also clarified the title of the section in the SI to "Calculation of the estimated BrC absorption lifetime and individual BrC chromophore lifetime"

"This procedure is used for individual chromophores as well as overall BrC absorbance."

Relating to the previous comment, there seems to be pretty high uncertainty in estimating lifetimes as this Figure S1 shows 1-2 data points and a linear regression. Please add a significant discussion on uncertainty bounds.

Please see the above comment about added error bars.

Is condensed phase filter-based? In the main text (page8 line 239), you refer to both filter-based and condensed phase. Could you please clarify which term refers to what?

We now explicitly state "filter-based" when contrasted with "solution phase" absorption spectroscopy measurements. Condensed phase is still used in the paper, but only used to describe the particulate matter photolysis.

---

## Author Comment (AC2) · 2 Nov 2019

**Reviewer #2**

We would like to thank the Reviewers for their constructive feedback. The reviewer comments are reproduced below in black, and our responses are provided below each comment in blue. We will also be uploading a track-changed version of the manuscript along with all responses as required by ACP.

1) The extraction procedure in the first paragraph of section 2.2 requires further elaboration. What is the rationale behind using the 3 organic solvents in these proportions? Is that based on an already established protocol? Have you verified that none of the BrC precipitates out of the water+DMSO mixture after evaporating the organic solvents? Why are there different extraction protocols for the fresh (water+DMSO) and photolyzed (only DMSO) particles?

An earlier publication (below) established that this organic solvent mixture resulted in the highest light absorption over the near UV and visible range, presumably due to a higher extraction efficiency. This is represented in Figure 1 of the paper. This paper is now cited.

Lin, P., Bluvshtein, N., Rudich, Y., Nizkorodov, S., Laskin, J., & Laskin, A. (2017). Molecular Chemistry of Atmospheric Brown Carbon Inferred from a Nationwide Biomass-Burning Event. Environmental Science & Technology, 51(20), 11561–11570. https://doi.org/10.1021/acs.est.7b02276

By visual inspection, there is not any precipitate from the water/DMSO mixture. However, there could be small, insoluble particles in the solution which we have not been able to see scattering from since the volume is very small (150 microliters). This is now clarified in the paper on page 6.

For the extracts of photolyzed extracts, the volume is much smaller at 30 microliters, and because many of the chromophores seem to dissolve better in DMSO than water, we simplified it by just adding DMSO. This does not affect the chromatography or the quality or the photodiode array spectra.

2) Line 235: Along the same lines, why is the extraction procedure for UV-vis different from HPLC/PDA/HRMS?

The extraction solvent mixture was altered depending on their advantages and disadvantages for the application: HPLC/PDA/HRMS or UV-Vis spectroscopy. Dichloromethane was avoided in the overall BrC photolysis experiments (UV-Vis experiments) and methanol was used instead. Dichloromethane absorbs strongly at <240 nm, and could interfere in the measurement of BrC absorption at near-UV wavelengths. In addition, from a safety standpoint, DCM is not a convenient solvent for absorption spectroscopy since it needs to be used in the hood due it its high volatility. Regardless of the solvent mixture used, it is impossible to extract all the material from filter with any solvent combination, and this is why the transmission spectroscopy measurements of the entire filter were done. The decay of absorption of visible wavelengths was consistent across both measurements (solution versus filter).

We now have explained the origins of the extraction protocol on page 6. An additional sentence on page 8 discusses why methanol was used in place of DCM.

3) The binning of the chromophores into Major, Intermediate, and Weak is informative, but can be confusing. I think it is helpful to stress that these categories involve the combined effects of abundance and absorption efficiency. For instance, an abundant chromophore with moderate

absorption efficiency can exist in the same bin as a less abundant chromophore with high absorption efficiency.

This is a good point. Our approach in this paper is to emphasize the observed contribution to the overall absorption by individual chromophores (rather than their relative concentration). Species such as nodakenetin ($C_{14}H_{14}O_4$) are a major contributor to the observed absorption for many of the fires. However we have not run standards to ascertain the emission factor for each compound and cannot determine the relative abundances of any of these compounds. This was the focus of Jen et al., 2018, which quantified the emission of particle phase compounds from the same fires, publishing the % contribution from each of the compound classes. To emphasize this point, we added the following sentence on page 9.

"Abundance and absorption cross section of BrC chromophores both factor into their assigned absorbance bin, as absorbance was not mass normalized with standards. It is possible that the chromophores labelled as "M" are present in small concentrations but have large absorption coefficient."

4) In addition to integrating the DPA signal over 300nm – 700 nm, can the authors also report wavelength-dependent information? Are there significant differences in AAE across the different chromophores? If yes, integrating absorption over say 4 sections (UV, short visible, mid visible, long visible) can provide interesting information. For example, a chromophore under category W but with say a relatively large AAE might be an I or an M in the UV range.

In general, BrC chromophores in this study are most absorbing in the UV and near UV, with some extending their absorption into the visible. While many have features in the near UV and visible with different $\lambda_{max}$ values, their relative integrated absorbance does not change all that much depending on the wavelength range (200-700 nm versus 300-700 nm), although many chromophores are excluded if only including visible wavelengths (400-700 nm). We have included absorption spectra in Tables 1 and 2 so that readers can check which chromophores are important in certain wavelength ranges (such as the long visible).

---

## Author Comment (AC3) · 2 Nov 2019

**Reviewer #3**

We would like to thank the Reviewers for their constructive feedback. The reviewer comments are reproduced below in black, and our responses are provided below each comment in blue. We will also be uploading a track-changed version of the manuscript along with all responses as required by ACP.

General comments

The Authors noted a decrease in absorption following photolysis that leveled off. This observation of a longer-lived fraction of BrC is consistent with several measurements in the lab (e.g. Wong et al. (2017)) and real environment (e.g. Forrister et al. (2015) and Di Lorenzo et al. (2017)). The previous work is not effectively reflected in the text of the paper. For example, lines 60-62 suggest that field measurements demonstrate a short lifetime for BrC. While observations of a short-lived fraction have been made, long-lived fractions have also been observed. In addition, the results described leading up to line 406 should be discussed in the context of this literature.

We have added to the introduction that these studies have observed a recalcitrant fraction of BrC, just as we did. This is in two places, on pages 3 and 4.

"However, there is a recalcitrant fraction of BrC that persists even after long aging times. Di Lorenzo et al. 2017 found that the fraction of higher molecular weight chromophores (>500 Da) relative to lower molecular weight chromophores (<500 Da) increased with plume transport time, on the order of hours to days."

"Size exclusion chromatography showed that low molecular weight BrC chromophores (<400 Da) were quickly formed and photodegraded giving yield to a photoenhancement due to the formation of high molecular weight species (>400 Da). They concluded that this high molecular weight fraction was responsible for long-lived light absorption."

We have also added a few sentences in the "Results and Discussion" section stating that other papers have observed the change in molecular composition of BrC chromophores, and this is consistent with our results.

"The results of both photolysis experiments is consistent with work by Di Lorenzo et al. (2017) and Wong et al. (2017) which show that during aging, high molecular weight BrC chromophores are formed after lower molecular weight chromophores are photodegraded. The high molecular weight fraction of BrC chromophores persist even at long aging times and are referred to as the recalcitrant fraction. This theory is one explanation for the short lifetimes of low molecular weight BrC compounds, while observing longer overall BrC absorption lifetimes."

More information should be provided about the solvents chosen for filter extraction. On line 139, the Authors say, "filters were extracted by solvents with a range of polarities". Only two solvent mixtures were used, and these were used to extract different samples.

This is still the introduction section and the purpose of this paragraph is to show the scope of the study and the main goals/results. We explain the choice of extraction solvents in the experimental section where we think it is more appropriate. See the below comment or details.

A mixture of dichloromethane/acetonitrile/hexanes was used to extract samples prior to HPLC/PDA/HRMS analysis, while a mixture of methanol/acetonitrile/hexane was used to extract

PTFE filters following transmission measurements. A justification should be provided for the choice of each solvent mixture. The impact of the solvent choice on the results described in lines 393-395 should also be discussed.

The extraction solvent mixture was altered depending on their advantages and disadvantages for the application: HPLC/PDA/HRMS or UV-Vis spectroscopy. Dichloromethane was avoided in the overall BrC photolysis experiments (UV-Vis experiments) and methanol was used instead. Dichloromethane absorbs strongly at <240 nm, and could interfere in the measurement of BrC absorption at near-UV wavelengths. In addition, from a safety standpoint, DCM is not a convenient solvent for absorption spectroscopy since it needs to be used in the hood due it its high volatility. Regardless of the solvent mixture used, it is impossible to extract all the material from filter with any solvent combination, and this is why the transmission spectroscopy measurements of the entire filter were done. The decay of absorption of visible wavelengths was consistent across both measurements (solution versus filter).

We now have explained the origins of the extraction protocol on page 6. An additional sentence on page 8 discusses why methanol was used in place of DCM.

Specific comments

Line 70: Typo in polycyclic aromatic hydrocarbons.

Thanks, we corrected this.

Line 122-123: Wong et al. showed not just a decrease in WSOC, but also a decrease in BrC absorption.

Indeed. We have clarified the text, which now reads:

"Wong et al. (2017) found that irradiated BBOA water extracts lost water soluble organic carbon (WSOC) when irradiated with 300-400 nm light. Simultaneously, the absorption coefficients at 365 nm and 400 nm first increased, in the latter case to about 0.035 $m^2\,g^{-1}$ after 20 minutes of illumination time, and then decreased to nearly zero in 60 minutes."

Line 266: Should be "one fewer methoxy ring substituent".

Done.

Line 318: Saying that PAHs are "stable during atmospheric transport" is an oversimplification. Although they may end up in pristine regions, the conditions that allow them to undergo long range transport are complex. The review from Keyte et al. (2013) and recent work from Zhou et al. (2019), along with other relevant work, should be consulted and discussed in this section.

Thank you. The text has been modified to clarify this point.

"Polycyclic aromatic hydrocarbons (PAHs) are known to be products of incomplete combustion, and they have the potential to be long-lived BrC chromophores despite their reactivity (Keyte et al., 2013). PAHs have been observed in pristine environments, and it has been suggested that this is due to phase separation of particles and slow diffusivity of PAHs to surfaces where they react with atmospheric oxidants (Fernández et al., 2002; Keyte et al., 2013; Macdonald et al., 2000; Sofowote et al., 2011; Zhou et al., 2012, 2019)."

Figure 1: Suggest this figure could be relocated to the SI.

We elected to keep this figure in the main paper for ease of reading since we reference it twice in the Results and Discussion.

---

## Author Response (AR2)

Response to 2nd review by Reviewer #3

Fleming et al. have carefully addressed the concerns of reviewers. I commend the authors on their thorough responses. They have addressed my concerns and, apart from a few minor comments/suggestions listed below, I believe this manuscript is suitable for publication in ACP. There is some inconsistency with using English names or molecular formulas for chemical species (e.g. line 62 "HONO and nitrogen dioxide").

We went through the manuscript trying to make names more consistent and refine writing in general. We did a number of other minor editing changes to improve the manuscript readability. The track-changed document is attached below for your reference.

My previous comment regarding the phrase (line 145) was not well expressed. My concern with this sentence is that it currently reads as if the same sample was extracted with solvents of multiple polarities (i.e. an extraction study), which this is not. I suggest re-phrasing slightly to clarify that different methods are applied to different samples.

[revised manuscript text omitted]
 t~T~heoretical s~S~tudy of a~A~queous *cis-*-p~P~inonic a~A~cid p~P~hotolysis, J. Phys. Chem. A, 117(48), 12930–12945, doi:10.1021/jp4093018, 2013.